# Oxidative Stress and Cellular Senescence Are Involved in the Aging Kidney

**DOI:** 10.3390/antiox11020301

**Published:** 2022-01-31

**Authors:** Laura Marquez-Exposito, Lucia Tejedor-Santamaria, Floris A. Valentijn, Antonio Tejera-Muñoz, Sandra Rayego-Mateos, Vanessa Marchant, Raul R. Rodrigues-Diez, Irene Rubio-Soto, Sebastiaan N. Knoppert, Alberto Ortiz, Adrian M. Ramos, Roel Goldschmeding, Marta Ruiz-Ortega

**Affiliations:** 1Cellular Biology in Renal Diseases Laboratory, IIS-Fundación Jiménez Díaz, Universidad Autónoma Madrid, 28040 Madrid, Spain; laura.marqueze@quironsalud.es (L.M.-E.); lucia.tejedor@quironsalud.es (L.T.-S.); antonio.tejera@quironsalud.es (A.T.-M.); srayego@fjd.es (S.R.-M.); vanessa.marchant@quironsalud.es (V.M.); irene.rubios@quironsalud.es (I.R.-S.); 2Red de Investigación Renal (REDinREN), Instituto de Salud Carlos III, 28040 Madrid, Spain; RRodriguez@fjd.es (R.R.R.-D.); aortiz@fjd.es (A.O.); AMRamos@fjd.es (A.M.R.); 3Department of Pathology, University Medical Center Utrecht, H04.312, Heidelberglaan 100, 3584 CX Utrecht, The Netherlands; F.A.Valentijn@umcutrecht.nl (F.A.V.); S.N.Knoppert-3@umcutrecht.nl (S.N.K.); R.Goldschmeding@umcutrecht.nl (R.G.); 4Translational Immunology Laboratory, Health Research Institute of Asturias (ISPA), Av. Roma, s/n, 33011 Oviedo, Spain; 5Division of Nephrology and Hypertension, IIS-Fundación Jiménez Díaz-Universidad Autónoma Madrid, 28040 Madrid, Spain

**Keywords:** cellular senescence, NRF2, aging kidney, oxidation, inflammaging, fibrosis

## Abstract

Chronic kidney disease (CKD) can be considered as a clinical model for premature aging. However, non-invasive biomarkers to detect early kidney damage and the onset of a senescent phenotype are lacking. Most of the preclinical senescence studies in aging have been done in very old mice. Furthermore, the precise characterization and over-time development of age-related senescence in the kidney remain unclear. To address these limitations, the age-related activation of cellular senescence-associated mechanisms and their correlation with early structural changes in the kidney were investigated in 3- to 18-month-old C57BL6 mice. Inflammatory cell infiltration was observed by 12 months, whereas tubular damage and collagen accumulation occurred later. Early activation of cellular-senescence-associated mechanisms was found in 12-month-old mice, characterized by activation of the DNA-damage-response (DDR), mainly in tubular cells; activation of the antioxidant NRF2 pathway; and klotho downregulation. However, induction of tubular-cell-cycle-arrest (CCA) and overexpression of renal senescent-associated secretory phenotype (SASP) components was only found in 18-month-old mice. In aging mice, both inflammation and oxidative stress (marked by elevated lipid peroxidation and NRF2 inactivation) remained increased. These findings support the hypothesis that prolonged DDR and CCA, loss of nephroprotective factors (klotho), and dysfunctional redox regulatory mechanisms (NRF2/antioxidant defense) can be early drivers of age-related kidney-damage progression.

## 1. Introduction

Chronic kidney disease (CKD) is emerging as an important health problem due to the absence of early diagnostic biomarkers and effective treatments. Although recent clinical trials have reported promising results with SGLT2 inhibition, most CKD patients still die prematurely or progress towards end-stage renal diseases (ESRD), needing kidney replacement therapies, such as dialysis or transplantation [1]. The aged population is constantly increasing, and kidney-aging is a risk factor for both acute kidney injury (AKI) and CKD [2]. In this regard, it is predicted that CKD will become the fifth-most-common global cause of death by 2040 [3]. Recent studies show that many features of aging characterize CKD, suggesting that CKD disease can be considered as a clinical presentation of premature aging [4,5]. Therefore, there is an urgent unmet medical need to understand the mechanisms of age-related kidney damage, as well as early biomarkers, to ensure better management of CKD.

Aging is related to telomere shortening due to replicative stress, and this can lead to cellular senescence [6]. This process is defined as an irreversible cell-cycle arrest (CCA), characterized by alterations in chromatin organization and gene expression, that induce profound phenotypic changes. Senescent cells possess a specific secretome known as senescence-associated secretory phenotype (SASP) [7,8], which is enriched with pro-inflammatory cytokines, growth factors, and profibrotic proteins [9,10]. Cellular senescence can be deleterious or beneficial depending on the biological context and timing. In this sense, the SASP can evoke a local inflammatory response with intricate and divergent effects, including the removal of senescent cells by phagocytosis, thus contributing to tissue remodeling and damage resolution [11,12,13], whereas persistent senescence-mediated inflammation leads to aberrant tissue remodeling and fibrosis [14]. In addition, the SASP, acting in an autocrine or paracrine manner in neighboring cells, can reinforce and propagate cellular senescence, by a process named secondary senescence [10].

Cellular senescence is induced in response to multiple types of damage, such as intense oncogenic signaling, DNA damage, telomere shortening, inflammation, oxidative stress, and toxins [9]. In the kidney, senescence has been involved in AKI, regeneration, AKI-to-CKD transition, CKD progression, transplant rejection, and aging [14]. The senescence mechanisms in the kidney include activation of the DNA damage response (DDR), which can be prolonged in time and activate the p53/p21cip1 axis and p16ink4a. Both p21cip1 (from now on p21) and p16ink4a (from now on p16) play a key role as inhibitors of cyclin-dependent kinases (CDK), blocking retinoblastoma tumor suppressor (Rb) phosphorylation mediated by CDKs, thus limiting cell proliferation and provoking CCA [14]. This triggers a cascade of phenotypic changes in tubular epithelial cells which release the aberrant SASP secretome, enriched in proinflammatory and profibrotic factors, aggravating kidney dysfunction and contributing to renal-damage progression [14,15,16]. Among the SASP proinflammatory components, several cytokines and chemokines, such as interleukin-6 (IL-6) and CCL-2 (MCP-1), could contribute to persistent renal inflammation and to further tubular cell injury and dysfunction [17,18] whereas SASP profibrotic factors, such as transforming growth factor-β (TGF-β) and cellular communication network factor 2 (CCN2/CTGF), could induce profibrotic responses, leading to kidney fibrosis [19,20].

Oxidative stress plays a key role in the premature aging associated with CKD [21]. Reactive oxygen species (ROS) accumulation and redox imbalance can lead to tissue damage due to increased breakage of the DNA and consequent DDR, which leads to CDK inhibitor activation [6,20]. This state of cells involves the activation of one of the molecular pathways to remove ROS, the nuclear factor (erythroid-derived 2)-related factor 2/hemooxigenase-1 (NRF2/HO-1) axis [9]. In physiological conditions, NRF2 is sequestered by Keap1, but upon a redox imbalance, NRF2 is released from Keap-1 and then translocated to the nucleus to bind to ARE sequences in DNA, initiating the transcription of genes related to antioxidant defense, such as HO-1 and NAD(P)H dehydrogenase (quinone 1) (NQO-1), among others [22]. Moreover, ROS and the NRF2/HO-1 pathway are involved in the increase and decrease, respectively, of inflammatory mediators [21]. In aging tissues, the NRF2/HO-1 pathway has been described to be dysfunctional, being unable to reduce either ROS production or inflammation [23]. Indeed, NRF2 impairment and sustained inflammation, characterized by increased infiltration of immune cells and overproduction of proinflammatory factors locally in the kidney, are relevant characteristics of CKD [21].

One important question in kidney research is the lack of non-invasive biomarkers to detect early kidney damage, including the onset of a senescence phenotype. Although p16 expression has been used as a classical senescence biomarker, there is not a specific biomarker to define the senescent cell state [14,24,25]. One recent study tried to identify robust senescence biomarkers in vivo, evaluating the mRNA expression profiles of a panel of known molecular hallmarks of senescence in multiple tissues in very aged (30 months) hybrid CB6F1 female mice, showing that p16 expression was upregulated in all tissues analyzed, including the kidney, whereas SASP components varied among tissues [26]. Despite this, the precise extent of senescent cell accumulation in aged animals, and a deeper characterization of age-related senescence in the kidney and its functional outcome remain unclear. Most of senescence-related experimental studies have been done in very old mice [27,28]. However, in a recent preclinical study we demonstrated an age-related increased susceptibility to develop more severe AKI in 12-month-old mice through exacerbation of senescence-related mechanisms [29]. Therefore, in this paper our aim was to investigate the age-related activation of cellular senescence, the redox-related mechanisms, and their correlation with early structural changes in the kidney, evaluating 3-, 12-, and 18-month-old C57BL6 mice.

## 2. Materials and Methods

### 2.1. Animals

Experiments were performed according to the European Community guidelines for animal experiments and the ARRIVE guidelines and with consent of the Experimental Animal Ethics Committee of the Health Research of the IIS-Fundación Jiménez Díaz and Proex065/18 of the Comunidad de Madrid. Six to nine C57BL/6 male mice per age group (3-, 12-, and 18-month-old) were studied.

All animals were sacrificed with an overdose of CO_2_ in a special chamber. Blood and urine were collected, and kidneys were perfused in situ with saline before removal. Half of each kidney (2/4) was fixed, embedded in paraffin, and used for immunohistochemistry, and the rest was snap-frozen in liquid nitrogen for renal cortex RNA and protein studies.

### 2.2. Protein Studies

Total proteins were isolated from frozen kidney tissue in lysis buffer as previously described [29] and quantified using a BCA protein assay kit (ThermoScientific; Waltham, MA, USA). Proteins (50 μg) were separated on 8–15% acrylamide gels using the SDS-PAGE, as described [29]. After electrophoresis, samples were transferred on to polyvinylidene difluoride membranes (Millipore; Burlington, VT, USA) blocked in TBS containing 0.1% Tween 20 and 5% dry non-fat milk for 1 h at room temperature and incubated in the same buffer with different primary antibodies overnight at 4 °C. After washing, membranes were incubated with the appropriate HRP (horseradish peroxidase)-conjugated secondary antibody (Invitrogen; Waltham, MA, USA) 1 h at room temperature and developed using an ECL kit (Amersham Biosciences; Piscataway, NJ, USA). Results were analyzed by LAS 4000 and Amersham Imager 600 (GE Healthcare; Chicago, IL, USA) and densitometered by Quantity One software (Bio-Rad laboratories; Hercules, CA, USA). The following primary antibodies were employed (dilution): NRF2 ((1:500); sc-365949, Sta. Cruz Biotechnology; Dallas, TX, USA), p21 ((1:500), Ab188224, Abcam; Cambridge, UK) and ERK1/2 ((1:500); sc-514302, Sta. Cruz Biotechnology).

### 2.3. Histology and Immunohistochemistry

Paraffin-embedded kidney sections were stained using standard histology procedures, as described elsewhere [29]. Periodic acid-Schiff- (PAS, Sigma-Aldrich; Burlington, VT, USA) stained slides were quantified, assessing tubular damage as tubular dilation and interstitial inflammatory infiltrate as arbitrary units as previously described [30]. Glomerular injury was determined by scoring of the morphological differences on PAS-stained slides. Severity of the glomerular injury was scored on a 0 to 5 scale wherein a kidney scored 0 when no glomerular injury was observed and a 5 when all capillary loops were replaced by amorphous weakly PAS-positive material. Picrosirius red staining was performed using a mixing of 1% Direct Red 80 (Sigma-Aldrich) and picric acid solution (Sigma), and slides were quantified using Image Pro-plus Software (Rockville, MD, USA) determining the positive red-staining area relative to the total area.

Immunohistochemistry (IH) was carried out in 3 μm thick tissue sections. The PTlink system (DAKO) was used for antigen-retrieving using sodium citrate buffer (10 mM) adjusted to pH 6–9, depending on the immunohistochemical marker. Endogenous peroxidase was blocked. Sections were incubated for 1 h at room temperature with 1X casein solution (Vector Laboratories) to remove non-specific protein-binding sites. Then, primary antibodies were incubated overnight at 4 °C. Specific HRP-conjugated (DAKO) or biotinylated secondary antibodies (Amersham Biosciences) were used. The latter were followed by avidin–biotin complex incubation (Vector Laboratories; Burlingame, CA, USA). Signal was developed with substrate solution and 3,3-diaminobenzidine as a chromogen (Abcam). Finally, slides were counterstained with Carazzi’s hematoxylin (Richard Allan Scientific; Kalamazoo, MI, USA). The primary antibodies used were: p21 ((1:2000), Ab188224, Abcam and γH2Ax (1:500), NB1002280 Novus Biological), F4/80 ((1:50), MCA497, Bio-Rad), myeloperoxidase ((1X), IS511, DAKO), 4-hydroxynonenal (HNE) ((1:1000), Ab46545, Abcam), and phosphorilated-NRF2 serine 40 ((1:2000), Ab76026, Abcam). Specificity was checked by omission of primary antibodies (not shown). Quantification was made by using the Image-Pro Plus software determining the positive-staining area relative to the total area or counting positive-staining manually (in the case of P21, γH2AX, myeloperoxidase, and F4/80 staining), in 5–10 randomly chosen fields (200× magnification).

### 2.4. Gene-Expression Studies

RNA from the renal cortex was isolated with TRItidy G^TM^ (PanReac; Barcelona, Spain). cDNA was synthesized by a high-capacity cDNA archive kit (Applied Biosystems; Waltham, MA, USA) using 2 μg total RNA primed with random hexamer primers following the manufacturer’s instructions. Quantitative gene expression analysis was performed on an AB7500 fast real-time PCR system (Applied Biosystems) using fluorogenic TaqMan MGB probes and primers designed by Assay-on-Demand^TM^ gene expression products. Mouse assays IDs were: *Cdkn1a*: Mm00432448_m, *Cdkn2a*: Mm00494449_m1, *Kl:* Mm00502002_m1, *Il6*: Mm00446190_m1, *Lcn2*: Mm01324470_m1, *Havcr1*: Mm00506686_m1, *Ctgf/Ccn2*: Mm01192933_g1, *Ccl2*: Mm00441242_m1, *Tgfb1*: Mm01178820_m1, *Hmox1*: Mm00516005_m1, *Nfe2l2*: Mm00477784_m1, *Serpine1*: Mm00435858_m1, *Cat:* Mm00437992_m1, and *Sod1*: Mm01344233_g1. Data were normalized to *Gapdh*: Mm99999915_g1 (Vic). The mRNA copy numbers were calculated for each sample by the instrument software using Ct value (“arithmetic fit point analysis for the lightcycler”). Results are expressed in copy numbers, calculated relative to 3-month-old mice group after normalization against *Gapdh.*

### 2.5. Statistical Analysis

Results are expressed as n-fold increase with respect to the average of 3-month-old mice as mean ± standard error of the mean (±SD), except for the PAS, p21, gH2AX, MPO, and F4/80 quantification which are expressed in arbitrary units. The Shapiro–Wilk test was used to evaluate sample normality distribution. If the samples followed the Gaussian distribution, a one-way ANOVA followed by the Fisher’s LSD test were used. To compare non-parametric samples, a Kruskal–Wallis was performed followed by the uncorrected Dunn’s test. Graphics and statistical analysis were conducted using GraphPad Prism 8.0 (GraphPad Software; San Diego, CA, USA). Values of *p* < 0.05 were considered statistically significant.

## 3. Results

### 3.1. Inflammation Precedes Tubular and Glomerular Lesions in Kidneys of Aged Mice

Kidney morphology was evaluated by PAS staining. Interstitial and perivascular inflammatory cell infiltration was observed in the kidneys of 12- and 18-month-old mice (Figure 1A). The inflammatory infiltration showed a significant increase in 12-month-old mice compared to young mice and was even higher at 18 months (Figure 1B). Glomerular damage, with presence of amorphous material and nodular expansion of the glomerular tuft, was observed in 18-month-old mice (Figure 1A,B). Despite the presence of inflammatory infiltrate in kidneys from 12-month-old mice, significant tubular dilation was not found until the age of 18 months, showing only a tendency to increase at 12 months (Figure 1A,B). Next, mRNA levels of the kidney injury biomarkers *Lcn2* (which encodes for NGAL) and *Havcr-1* (which encodes for KIM-1) were evaluated [31,32,33,34,35]. Both genes were upregulated only at 18 months (Figure 1C). Renal function, assessed by serum creatinine, BUN and urea, was only found deregulated in 18-month-old mice (Figure 1D–F).

Kidney infiltrating cells were further characterized by immunohistochemistry using specific markers for neutrophils (myeloperoxidase, MPO) and macrophages (F4/80-positive cells). Infiltration by neutrophils and monocytes/macrophages was observed in the kidney cortex of 12- and 18- month-old mice, not finding substantial differences between aged mice in MPO, but showing higher F4/80-positive infiltrating cells in 18-month-old mice (Figure 2). Taken together, the spontaneous development of tubular and glomerular lesions in naturally aging mice was preceded by kidney inflammatory infiltration.

### 3.2. Aged Kidneys Display Increased Collagen Accumulation

To determine whether aged kidneys presented aberrant extracellular matrix accumulation (ECM) a picrosirius red staining was performed and collagen content was evaluated. Kidney cortex showed a slight increase in collagen accumulation by 12 months, but only in 18-month-old mice was collagen deposition significantly higher than in 3-month-old mice (Figure 3). Thus, 18-month-old mice developed kidney fibrosis.

### 3.3. Activation of Senescent Mechanisms Associated with DNA-Damage Response (DDR) Precedes Cell-Cycle Arrest (CCA) and Induction of an Aberrant Secretome (SASP)

To evaluate the development of a senescence phenotype in the aging kidney, several markers of senescence-associated mechanisms were assayed. First, activation of DDR was evaluated using the DDR marker γH2AX by IH. In kidneys of 12-month-old mice, γH2AX-positive nuclei were found, mainly in tubular cells and in some infiltrating immune cells, whereas almost no γH2AX-positive nuclei were observed in 3-month-old mice. Similar levels of γH2AX-positive nuclei were observed in 12- and 18- month-old mice (Figure 4A,B).

Next, CCA changes were investigated evaluating p21 at gene and protein levels. Although a slight increase in *Cdkn1a* mRNA levels were observed in the kidneys of 12-month-old mice, only at 18 months were gene and protein levels significantly upregulated compared to 3-month-old mice (Figure 4A,C,D,F). Using IH we found several p21-positive nuclei in both tubular and interstitial cells in the kidney of aging mice (Figure 4A). The CDK inhibitor p16 was also explored. Although a tendency for higher *Cdkn2a* mRNA levels was observed in 12-month-old mice, a significant upregulation of *Cdkn2a* mRNA levels was only observed in 18-month-old mice (Figure 4E).

Another feature of senescent cells is an SASP, including the increased production of proinflammatory and profibrotic factors [15,25]. Analysis of the gene-expression levels of the proinflammatory SASP components *Ccl2*, *Il6,* and *Il1b* and the profibrotic factors *Tgfb1, Ccn2/Ctgf,* and *Serpine1* (which encodes PAI-1) showed that all of them were only upregulated in 18-month-old mice (Figure 5), but not in 12-month-old mice. Taken together, these data indicate that aging-associated senescent-cell accumulation involves initial DDR as the first mechanism of damage, subsequently followed by CCA and SASP.

### 3.4. The Anti-Aging Factor Klotho Is Lost Early during Renal Aging

Klotho is rapidly downregulated when kidneys are injured, making this factor a marker of early kidney damage, and associated with the aging and cellular senescence in the kidney [29,36,37,38,39]. As expected, *klotho* mRNA levels were downregulated in 12-month-old mice, and a significant loss of klotho was found in 18-month-old mice when compared to 12-month-old mice (Figure 6), suggesting an over-time progressive aging-associated loss of klotho.

### 3.5. NRF2 Pathway Is Deregulated in the Aging Kidney

The timeline of the NRF2 pathway was evaluated over time at kidney gene, protein, and activation levels. Both *Nfe2l2* (the gene that encodes for the NRF2 protein) gene expression and NRF2 total protein expression levels were elevated in the kidneys from 18-month-old mice, but not from 12-month-old mice, both comparing to those of 3-month-old mice (Figure 7A,B). Moreover, the activation of the NRF2 pathway was explored by evaluating NRF2 phosphorylation by IH. Interestingly, IH quantification showed that NRF2 overactivation peaked at 12 months as assessed by increased phosphorylated NRF2 in cell nuclei. NRF2 activation diminished thereafter, and phosphorylated-NRF2 levels were similar in 18- and 3-month-old mice (Figure 7). These data showed an early activation of the NRF2 pathway, observed at 12 months, followed by an upregulation of NRF2 protein levels associated with increased de novo gene expression, but not associated with sustained activation of the pathway.

### 3.6. A Progressive Redox Imbalance Is Established in the Aging Kidney

Oxidative stress is another mechanism associated with senescence and kidney damage [6,20,21]. Gene expression of redox-response-related factors, including *Cat*, *Sod1,* and *Hmox-1* (which encodes for catalase, superoxide dismutase-1, and HO-1 protein, respectively) were analyzed. Renal gene expression of *Cat* was downregulated at 12 months compared to 3 months, and further decreased at 18 months (Figure 8), whereas *Sod1* gene expression was only diminished at 18 months. In contrast, *Hmox1* mRNA levels were elevated in the kidneys from 18-month-old mice, but not at earlier time points (Figure 8).

Finally, lipid peroxidation, one of the final responses of oxidation in CKD [40,41], was studied by staining for 4-HNE. A significant increase of lipid peroxidation was only observed in the kidneys of 18-month-old mice, but not at 12-months-old (Figure 9).

## 4. Discussion

Our findings support the hypothesis that DDR activation, loss of protective factors (klotho), and dysfunctional redox regulatory mechanisms can be early drivers of age-related induction of senescence mechanisms leading to similar features of kidney damage progression (Figure 10), and supporting that CKD could be a model of accelerated aging.

In humans, glomerular filtration rate (GFR) decreases progressively, starting from age 18–24 years [42]. At the age of 50–60 years, even healthy human kidneys suffer macrostructural changes, such as a decrease in the cortex and increase in the medullary volume, increasing the surface roughness and the number of cysts. Moreover, nephron loss is directly related to GFR decline, and nephrons are lost with aging [43], but there is no clear relation between activation of senescence mechanisms and loss of kidney function in healthy humans. Although many experimental studies have investigated the activation of senescence mechanisms in response to kidney injury, little attention has been given to the characterization of the activation of senescence in healthy mice, and most of the studies have been done in very old mice [44,45,46]. Our studies demonstrate that 12-month-old C57Bl6 mice present an early activation of pro-senescence mechanisms in tubular cells, characterized by high nuclear γH2AX in some tubular epithelial cells, suggesting DNA damage and subsequent DDR activation. DDR markers can activate p21/p53 downstream pathways to induce cellular senescence [14,25]. However, in 12-month-old mice, the presence of p21-positive cells in the kidney was scarce and renal mRNA levels were slightly, but not significantly upregulated. Interestingly, in 18-month-old mice, an increase in p21-positive tubular cells, increased p21 and p16 gene expression levels, and overexpression of SASP genes were found. Furthermore, this was associated with tubular damage and fibrosis, exclusively in 18-month-old mice. These data show that at the age of 12 months, the mouse kidney exerts DDR activation, but at this age is not associated with tubular damage and fibrosis. However, the deleterious age-related changes continue with time. At 18 months, tubular cells presented positive senescence markers (nuclear γH2AX and p21 expression), associated with tubular damage and collagen accumulation (Figure 10). Accordingly, in a study done in CB6F1 hybrid female mice, a strain reported to have an average lifespan of 30 months [47] which is considerably longer than the average lifespan of 25 months reported for C57BL/6mice [48], p21 gene expression was increased at 12 months, whereas p16 only increased after 24 months [49]. In previous studies in aged kidneys from mice and humans with allograft nephropathy, p16 and p21 expression were highly correlated with structural and functional histological changes [50,51]. These studies support our data, since we observed tubular damage and peak expression of p16 and p21 only in 18-month-old mice, indicating that these cellular senescence markers could represent a late stage of dysfunction when the kidney damage is already established. Morphological changes and loss of mitochondrial function would lead to ROS production, triggering oxidative stress and therefore accelerating the progression of renal fibrosis [49]. In a previous study in C57BL/6 mice, mitochondrial dysfunction was only found at 24 months, but not earlier [49]. Our findings showing a tendency for increased collagen accumulation at the earliest time points suggest that other mechanisms, besides mitochondrial dysfunction, could be involved in age-related kidney fibrosis.

SASP-mediated secondary senescence can be another mechanism of age-related kidney damage, and could contribute to kidney-damage progression, causing sustained inflammation and fibrosis. In this sense, 18-month-old murine kidneys presented a significant upregulation of SASP components, including profibrotic factors, such as TGF-β and CCN2, and maintained the γH2AX and p21/p16 activation in tubular-epithelial cells, contributing to amplifying tubular damage by secondary senescence and fibrosis. The release of SASP components by senescence cells can affect the growth, migration, and differentiation of neighboring cells, mainly impacting overall tissue architecture, and promoting chronic inflammation [52,53]. Changes in SASP components have been previous investigated in very old (30 months) female mice showing an upregulation of proinflammatory factors *Il1b, ccl8, Cxcl1,* and *Cxcl2* in kidneys [26]. Moreover, SASP factors can contribute to NF-κB pathway activation, which is known to have a relevant role in establishing and maintaining SASP component production and DDR activation [54]. Taken together, both the DDR and SASP are involved in releasing more pro-senescence secretome factors, thus amplifying the inflammatory and fibrotic response which is one of the mechanisms involved in the AKI-to-CKD transition and in CKD progression [54].

Aging has been related to low-grade chronic inflammation, in a process termed “inflammaging”, in which both the innate and acquired immune responses are dysregulated [55,56,57]. Human kidney transcriptomics disclosed evidence of inflammaging in aging kidneys [58]. Moreover, healthy and transplanted aged human kidneys have higher inflammatory infiltration than young ones [59]. Consistently, we have found the presence of inflammatory cell infiltration, including macrophages and neutrophils, in the kidneys from 12-month-old mice, but at this point there were no significant changes in kidney gene expression levels for several proinflammatory factors, including SASP components or chemokines, that could be responsible for immune recruitment into the kidney. Importantly, “immunosenescence” has been described as the decline of immune-cell efficiency due to aging [60]. Immunosenescent cells differ from healthy immune cells, as they express more proinflammatory factors, display different the CD membrane expression markers (for T cells, overexpression of CD57, and loss of CD28), and express CCA proteins such as γH2AX, p21, and p16 [61] and an altered secretome [60]. Immunosenescent cells accumulate in different tissues during natural aging [60]. In our study, we have observed increased inflammatory cell infiltration associated with the presence of γH2AX- and p21-positive interstitial cells in 18-month-old kidneys. Although future studies are needed to clearly demonstrate that the inflammatory cell infiltration exert phenotypic senescent-related changes, these data show the importance of the study of the role of immunosenescent inflammatory cells in kidney-damage progression. Experimental studies have demonstrated that the selective, senolytic-mediated elimination of senescent cells or the disruptions of the SASP program could be used as potential therapeutic strategies against aging [62]. All these data show the complexity of senescence regulation and functional consequences, indicating the necessity for further research in this area.

Oxidative stress is one of the possible inducers of senescence, since ROS production can damage the DNA and activate the DDR [16]. Moreover, prolonged ROS production activates the NRF2/antioxidant responsive element (ARE) pathway, implicated in detoxification, through NRF2 phosphorylation and translocation to the nucleus, leading to transcription regulation of NRF2 target genes. Several of these genes are involved in the antioxidant response, such as *Cat*, *Sod-1*, and *Hmox-1* among others [63,64,65]. Some evidence shows that cellular senescence is directly associated with NRF2 pathway impairment, since overactivation of the NRF2 pathway decreases the expression of some senescence markers, such as p21 [63], whereas NRF2 inhibition produces cellular senescence [64]. Different preclinical studies have demonstrated an impairment of the NRF2 antioxidant pathway in CKD. In a model of murine subtotal nephrectomy, NRF2 nuclear levels and the expression of its target genes *Hmox-1*, *Cat,* and *Gpx4*, were decreased in injured kidneys [66]. In the model of unilateral ureteral obstruction, NRF2 was initially overactivated after the injury, but downregulated in the chronic phase of the disease, showing that NRF2-pathway inactivation is associated with sustained inflammation and damage progression [21]. Our data support these findings, since NRF2 is overactivated in 12-month-old mice, indicating a redox imbalance and highlighting that mild damage is already occurring in the kidney. However, at later time points, the NRF2 pathway is impaired, as evidenced by the downregulation of active NRF2 and decreased *Cat* and *Sod-1* gene expression in 18-month-old mice. Of note, the observed upregulation of mRNA and total NRF2 protein levels at this time-point could represent a compensatory mechanism, but this pathway is already dysfunctional (Figure 10). On the other hand, NRF2 activation goes beyond cytoprotective properties, as it is implicated in lipid metabolism [67]. In CKD, aberrant quantities of plasma lipids are a target of ROS, creating subproducts such as 4-HNE, among others [40]. In our study, we only observed 4-HNE production at 18 months, suggesting an excess of ROS production and lipid deposition, not compensated by efficient antioxidant responses, as evidenced by low *Cat* and *Sod-1* gene expression. These data indicate that 18-month-old mice present features of CKD, as NFR2 impairment and oxidative stress, not observed in 12-month-old mice (Figure 10). All these findings suggest that targeting the NRF2 pathway should be explored as a potential therapy for kidney aging.

Another remarkable factor involved with ROS production and cellular senescence is the nephroprotective hormone klotho [29,49]. which was downregulated in kidneys from 12-month-old mice (Figure 10). Klotho is normally expressed and secreted by tubular cells and has anti-aging, anti-inflammatory, and anti-fibrotic properties [68,69,70]. For example, klotho protects endothelial cells from senescence [71], and is a marker of aging and cellular senescence [72,73]. Klotho downregulation can be both a consequence and driver of inflammaging and increased ROS production in kidney disease [36,38,39,50,74]. The increase in ROS production due to a klotho downregulation could explain the overactivation of NRF2 in 12-month-old mice, as in these mice we have already found low levels of klotho. Despite NRF2 activation, in 18-month-old mice, klotho is further downregulated, NRF2 activation impaired, and pro-fibrotic gene expression and collagen deposition highly increased, in line with the role of klotho in diminishing cellular senescence and renal fibrosis, the hallmark of CKD [75].

## 5. Conclusions

Our preclinical studies in C57BL/6 mice describe early age-related changes in 12-month-old mouse kidneys characterized by the loss of the nephroprotective factor klotho, the activation of several protective responses, such as DDR and NRF2/antioxidant defense, and the presence of infiltrating cells. At this point in time, the investigation of biomarkers of early damage could be very interesting. However, the deleterious age-related changes progress over time, since in 18-month-old mice, tubular damage and kidney fibrosis were already associated with tubular-senescence phenotype changes, including cell-growth arrest and SASP overexpression, supporting the hypothesis of senescent cells as drivers of age-related kidney-damage progression (Figure 10). These processes present many similarities with mechanisms involved in CKD progression and support further research in this area to prevent kidney damage even in healthy populations.

## Figures and Tables

**Figure 1 antioxidants-11-00301-f001:**
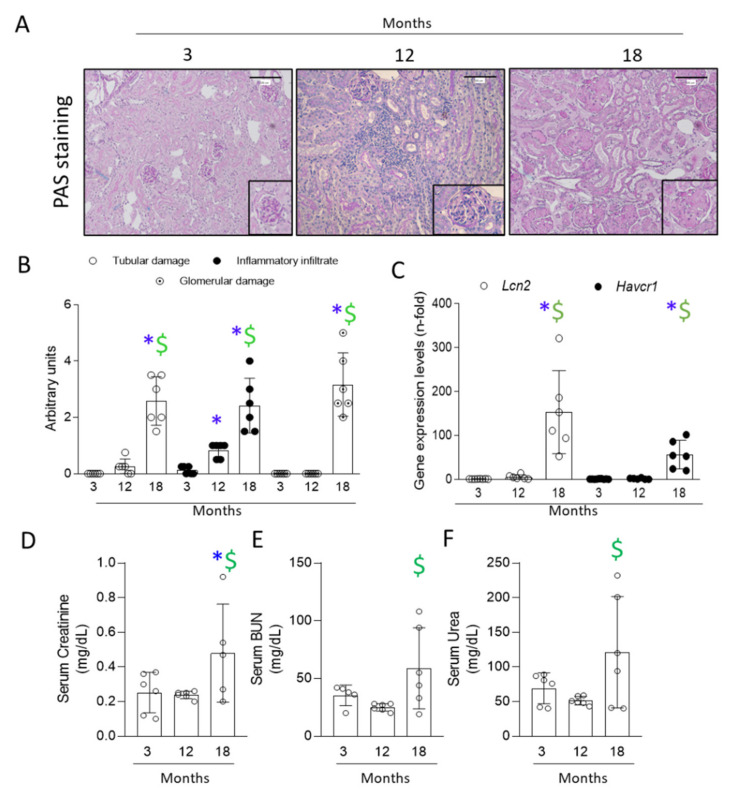
Glomerular and tubular damage, as well as changes in renal function parameters were only found in 18-month-old mice. Kidneys from 3-, 12- and 18-month-old C57BL/6 mice were studied. (**A**) Representative PAS staining microphotographs per group at 200× magnification, and glomeruli detail. Scale Bar: 100 μm. (**B**) PAS staining score (from 0 to 4) was categorized as tubular damage (defined as tubular dilatation and tubular atrophy) or inflammatory infiltrate (interstitial and perivascular cells). Glomerular capillary loops replaced by PAS-positive material were scored from 0 to 5. Data are presented as arbitrary units. (**C**) qRT-PCR from kidney extracts analyzing *Lcn2* (which encodes NGAL) and *Havcr1* (which encodes KIM-1). Data are represented as n-fold. (**D**) Serum creatinine, (**E**) BUN, and (**F**) urea, represented in mg/dL. All data are expressed as mean ± SD of 6–9 animals per group. * *p* < 0.05 vs. 3-month-old mice and $ *p* < 0.05 vs. 12-month-old mice. The non-parametric Kruskal–Wallis statistical test followed by the uncorrected Dunn’s test was performed.

**Figure 2 antioxidants-11-00301-f002:**
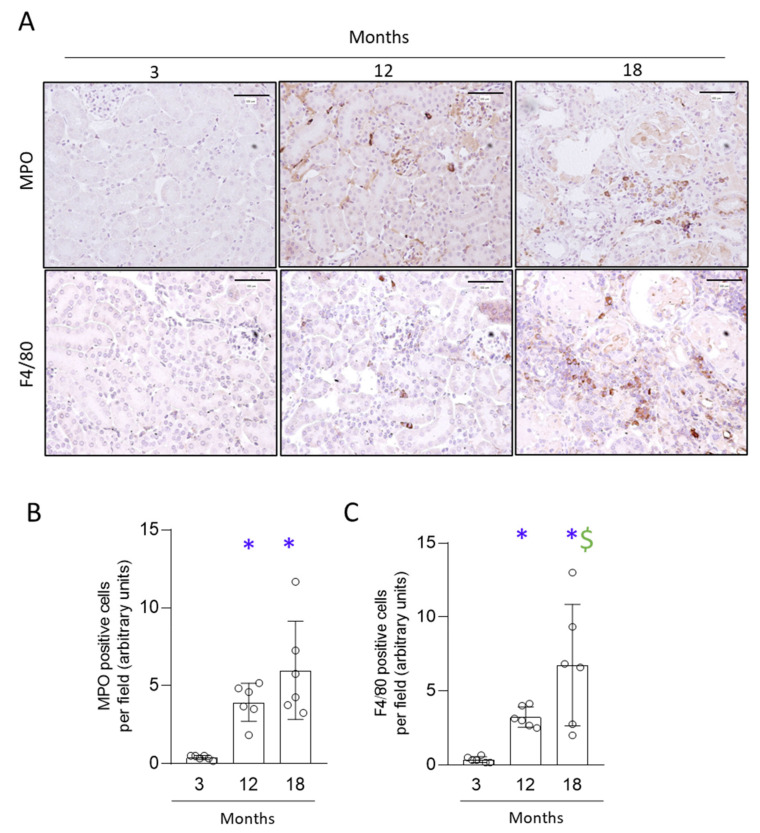
Interstitial inflammatory cell infiltration was already observed in 12-month-old mice and maintained in older mice. Kidneys from 3-, 12- and 18-month-old C57BL/6 mice were stained for myeloperoxidase (MPO) and F4/80 by immunohistochemistry. (**A**) Representative microphotographs of MPO (neutrophil marker) and F4/80 (macrophage and dendritic cell marker) at 200× magnification. Scale Bar: 100 μm. (**B**) MPO quantification and (**C**) F4/80 quantification of the average of positive cells per field, presented as arbitrary units and expressed as mean ± SD of six animals per group. * *p* < 0.05 vs. 3-month-old mice and $ *p* < 0.05 vs. 12-month-old mice. The one-way ANOVA statistical test followed by the Fisher’s LSD test was performed.

**Figure 3 antioxidants-11-00301-f003:**
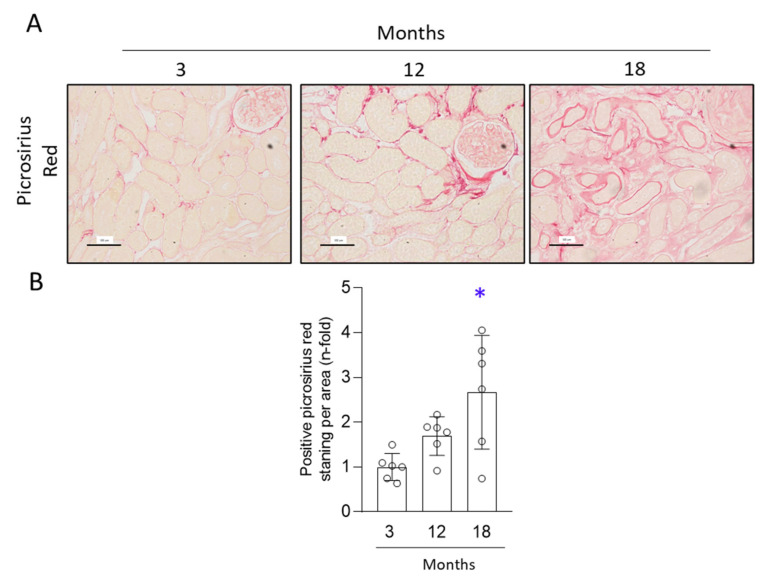
Collagen accumulation was found in the kidneys of 18-month-old mice. Paraffin-embedded kidneys from 3-, 12- and 18-month-old C57BL/6 mice were stained with picrosirius red. (**A**) Representative microphotographs of collagen deposition assessed by picrosirius red staining at 200× magnification. Scale Bar: 100 μm. (**B**) Positive picrosirius red staining quantification per total area, presented as n-fold and expressed as mean ± SD of six animals per group. * *p* < 0.05 vs. 3-month-old. The one-way ANOVA statistical test followed by the Fisher’s LSD test was performed.

**Figure 4 antioxidants-11-00301-f004:**
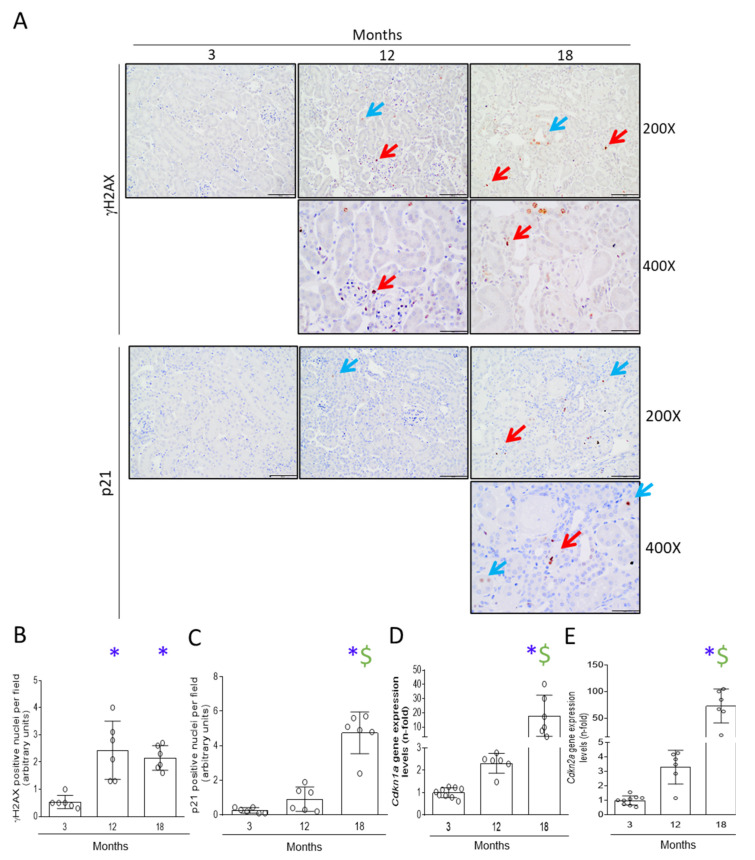
Early activation of DNA damage response (DDR) in 12-month-old mice kidneys was followed by induction of cell-cycle arrest (CCA) in 18-month-old kidneys. Kidneys from 3-, 12- and 18-month-old C57BL/6 mice were studied. (**A**) Representative immunohistochemistry microphotographs for γH2AX, a DDR marker, and p21, a cyclin-dependent kinase inhibitor which is a CCA marker, at 200× (scale bar: 100 μm) and at 400X (scale bar: 50 μm) magnification. Red arrows mark positive interstitial nuclei and blue arrows mark positive tubular epithelial cell nuclei. (**B**,**C**) show quantification of γH2AX- and p21-positive nuclei per field, respectively. Data are presented as arbitrary units. qRT-PCR from kidney extracts for (**D**) *Cdkn1a* (which encodes for p21) and (**E**) *Cdkn2a* (which encodes for p16) markers of CCA and cellular senescence. (**F**) Quantification of p21 protein levels by Western blot in the upper panel and the representative blots in the lower panel, using ERK2 protein levels as loading control. Data are presented as n-fold and expressed as mean ± SD of 6–9 animals per group. * *p* < 0.05 vs. 3-month-old mice and $ *p* < 0.05 vs. 12-month-old mice. The non-parametric Kruskal–Wallis statistical test followed by the uncorrected Dunn’s test was performed.

**Figure 5 antioxidants-11-00301-f005:**
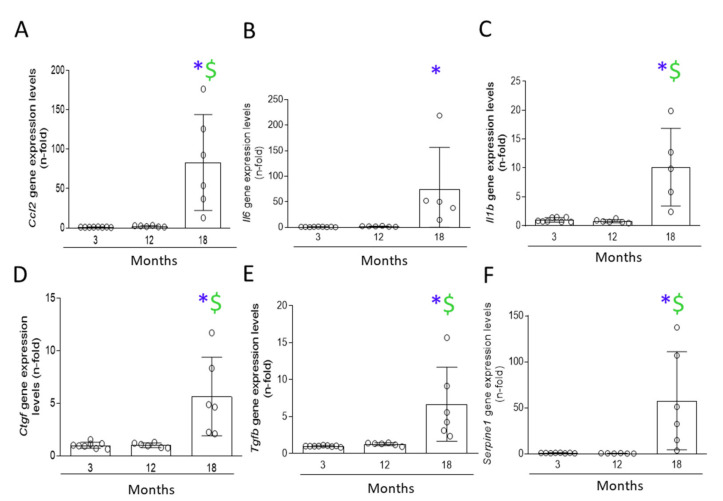
Senescence-associated secretory phenotype (SASP) was found in 18-month-old mice kidneys. Kidneys from 3-, 12-, and 18-month-old C57BL/6 mice were studied. (**A**) *Ccl2,* (**B**) *Il6,* (**C**) *Il1b,* (**D**) *Ctgf,* (**E**) *Tgfb1,* and (**F**) *Serpine1* were analyzed in kidney tissue by qRT-PCR. Data are presented as n-fold and expressed as mean ± SD of 6–9 animals per group. * *p* < 0.05 vs. 3-month-old mice and $ *p* < 0.05 vs. 12-month-old mice. The one-way ANOVA statistical test followed by the Fisher’s LSD test was performed.

**Figure 6 antioxidants-11-00301-f006:**
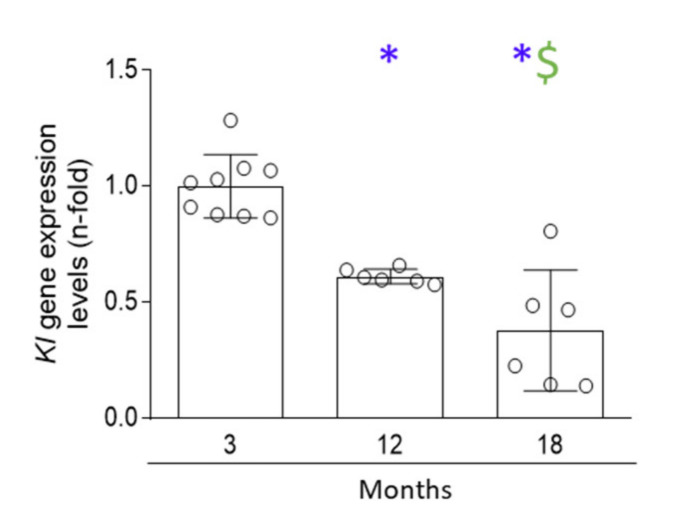
Klotho gene expression levels showed a marked decline in 12-month-old mice which was more downregulated in 18-month-old mice. In kidneys from 3-, 12-, and 18-month-old C57BL/6 mice, klotho gene expression was studied by qRT-PCR. Data are presented as n-fold and expressed as mean ± SD of 6–9 animals per group. * *p* < 0.05 vs. 3-month-old mice and $ *p* < 0.05 vs. 12-month-old mice. The one-way ANOVA statistical test followed by the Fisher’s LSD test was performed.

**Figure 7 antioxidants-11-00301-f007:**
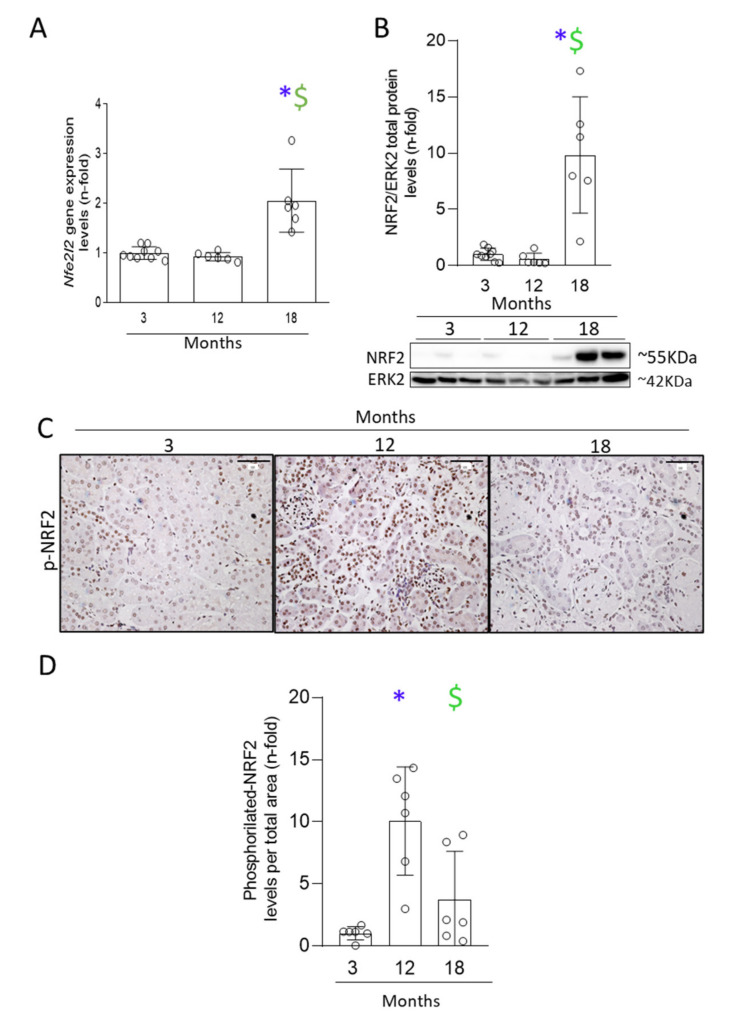
Early activation of antioxidant NRF2 pathway in 12-month-old mice was followed by a deregulation in 18-month-old mice kidneys. Kidneys from 3-, 12-, and 18-month-old C57BL/6 mice were studied. (**A**) qRT-PCR from kidney extracts of *Nfe2l2* gene expression levels. (**B**) Quantification of NRF2 protein levels by Western blot in the upper panel and the representative blots in the lower panel, using ERK2 protein levels as loading control. (**C**) Quantification of nuclei stained for phosphorylated-NRF2 per total area and (**D**) representative microphotographs at 200× magnification. Scale bar: 100 μm. Data are presented as n-fold and expressed as mean ± SD of 6–9 animals per group. * *p* < 0.05 vs. 3-month-old mice and $ *p* < 0.05 vs. 12-month-old mice. The non-parametric Kruskal–Wallis statistical test followed by the uncorrected Dunn’s test was performed.

**Figure 8 antioxidants-11-00301-f008:**
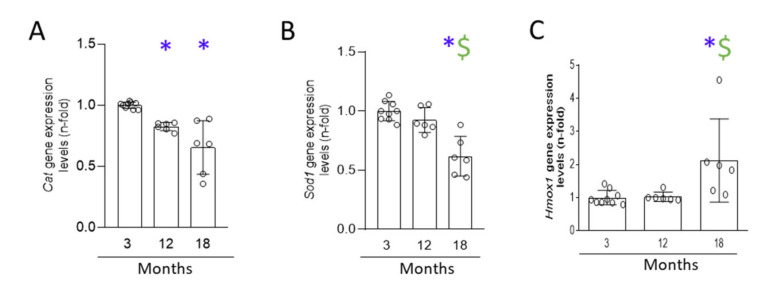
Changes in NRF2 target genes during murine kidney aging. Kidneys from 3-, 12-, and 18-month-old C57BL/6 mice were studied. (**A**) *Cat,* (**B**) *Sod-1,* and (**C**) *Hmox-1* gene expression was analyzed by qRT-PCR of kidney tissue. Data are presented as n-fold and expressed as mean ± SD of 6–9 animals per group. * *p* < 0.05 vs. 3-month-old mice and $ *p* < 0.05 vs. 12-month-old mice. The non-parametric Kruskal–Wallis statistical test followed by the uncorrected Dunn’s test was performed.

**Figure 9 antioxidants-11-00301-f009:**
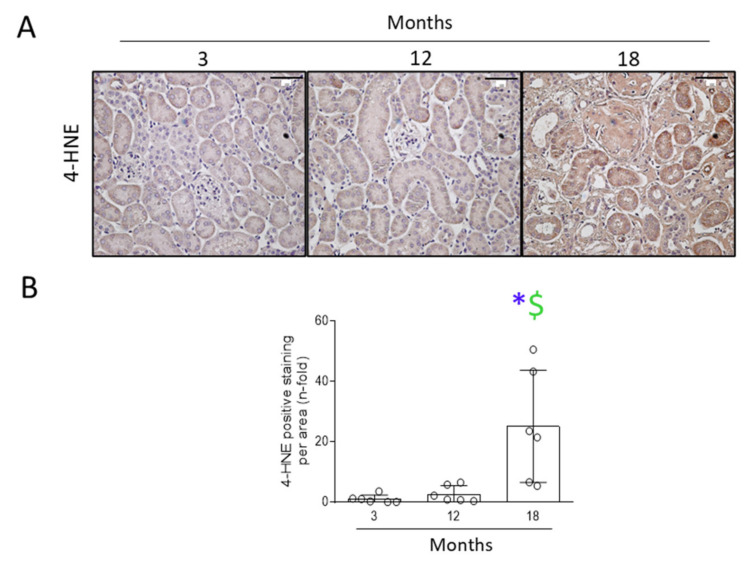
The lipid peroxidation marker 4-hydroxinonenal was increased in 18-month-old mouse kidneys. Kidneys from 3-, 12-, and 18-month-old C57BL/6 mice were extracted and studied. (**A**) Representative microphotographs of the 4-hydroxinonenal (4-HNE) immunohistochemistry at 200× magnification. Scale bar: 100 μm. (**B**) Quantification of 4-HNE protein levels per total area. Data are represented as n-fold and expressed as mean ± SD of six animals per group. * *p* < 0.05 vs. 3-month-old mice and $ *p* < 0.05 vs. 12-month-old mice. The non-parametric Kruskal–Wallis statistical test followed by the uncorrected Dunn’s test was performed.

**Figure 10 antioxidants-11-00301-f010:**
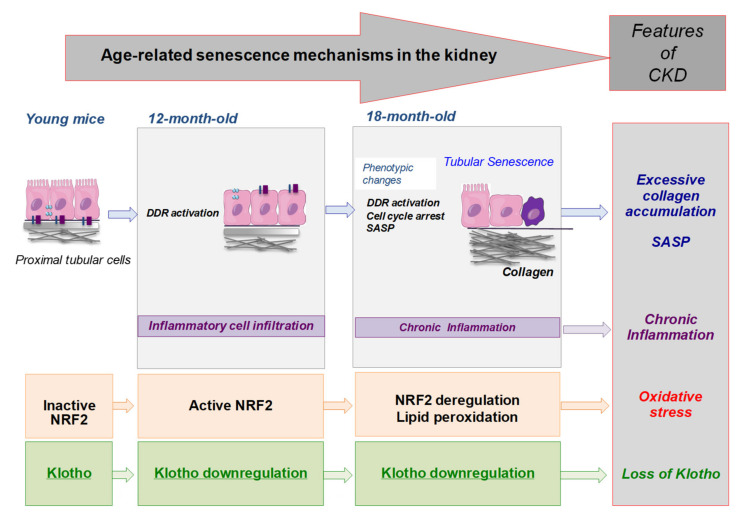
Conceptual representation of age-associated mechanisms in murine kidneys. The earliest changes in aging kidneys include the activation of pro-senescent responses, such as DDR in proximal tubular cells, the appearance of infiltrating inflammatory cells, the activation of NRF2 antioxidant defense, and the loss of the nephroprotective factor klotho before morphological changes occur. In the long-term, DDR activation is maintained and is a driver of cell-cycle-arrest-induced cellular senescence, which is observed in tubular cells. Moreover, there is an induction of SASP, sustained inflammatory cell infiltration, accumulation of collagen, and deregulation of NRF2 pathway, associated with increased oxidative stress. All of these are hallmarks of CKD.

## Data Availability

Data is contained within the article.

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
