# Peer review of "Oxidative Stress and Cellular Senescence Are Involved in the Aging Kidney"

_antioxidants, 2022, doi:10.3390/antiox11020301_

Round 1
Reviewer 1 Report
The manuscript presented by Marquez-Esposito et al. is of sure interest, providing new findings in the comprehension of possible mechanisms leading to renal senescence.
Authors reported, in animal model, that mice aging is correlated with renal loss of Klotho factor (supposed as nephroprotective), and with the activation of several protective responses, such as DDR and NRF2/antioxidant defense, and the presence of infiltrating immunosenescent cells at 12 months. At 18 months tubular damage and kidney fibrosis was already associated with tubular senescence phenotype changes including cell growth arrest and SAPS overexpression, supporting the hypothesis of senescent cells as drivers of age-related kidney damage progression.
Despite the relevant findings, I present some concerns:
- as well reported in Discussion, in healthy humans, glomerular filtration rate decreasing is the hallmark of kidney aging. Authors did not provide any evaluations of glomerular compartments, focusing only on tubular cells. May authors provide histological analysis of glomeruli?
- could authors provide ACR and/or serum creatinine or serum BUN? To correlate histological findings with classical renal biomarkers
Author Response
See the answer in the attached PDF file. Thank you.

Reviewer 2 Report
The authors investigated the age-related activation of cellular senescence-associated changes in the kidney of 3-, 12-, and 18-month-old C57BL/6 mice. Inflammatory white blood cells (neutrophils and macrophages) infiltration (Figures1 and 2) and fibrosis-related collagen accumulation (Figure 3) were observed in the kidney of 12-month-old mice and increased further at the later (18 months). Along with those, DNA damage response (DDR, by phosphorylation of gamma-H2AX) (Figure 4A upper) and decrease in Kl expression (Figure 6) were also observed from 12-month-old mice and progressed further at the later. On the other hand, tubular damages (Figure 1B), increases in the expression of kidney injury markers (Lcn and Havcr1, Figure 1C), cell cycle arrest (CCA) markers (Cdkn1A and Cdkn2A, Figures 4E and 4F), and senescence-associated secretory phenotype-related genes (Ccl2, Il6, Il1b, Ctgf, Tgfb1, and Serpine1, Figure 5), and accumulation of the lipid peroxidation (Figure 9) in the kidney were observed only in 18-month-old mice. NRF2 pathway was activated at 12 months (by phosphorylation at Ser 40, Figures 7C and 7D) and dysregulated at 18 months (overexpression and increase in the expression of the downstream genes, Figures 7A, 7B, and 8). These detailed observations support the authors’ hypothesis that prolonged DDR and CCA, loss of Kl, dysfunction of NRF2 pathway, and infiltrating immunosenescent cells can be early drivers of age-related kidney damage progression.
The results of this study must be exciting and informative for the investigators engaged in chronic kidney disease. However, the manuscript has unclear things and issues to be revised in the text as described below.
Major issues:
The authors intend to show that immunosenescence due to aging engages in the process of CKD in old mice. However, they did not show the existence of immunosenescent cells at all. Furthermore, although inflammatory infiltration of neutrophils and macrophages in the kidney were observed at 12-month-old and increased age-dependently, this phenomenon is independent of immunosenescence. Therefore, their experimental results do not support the description in lines 416-425 in the discussion section. The gammaH2AX- and p21-positive cells in the kidney are probably kidney cells. Is it natural to think that DDR-positive kidney cells increase during aging, the resulting damaged cells recruit the interstitial inflammatory infiltration, and these inflammatory white blood cells injure kidney cells? The reviewer does not think their results support the conceptual representation that senescent immune cells are involved in CKD progress in Figure 10, either. Please remove the part of immunosenescent cells from Figure 10. Please rewrite the image Figure 10 in compliance with experimental results in this study.
Minor
- Line 93, the term “inflammaging” should be explained. The word appeared very recently.
- Lines 180-189, standard derivation (SD) should be used instead of SE. SE should be used when the same experiments are performed repeatedly. Although they used six to nine mice per age group, they did not repeat the same experiments for several times.
- Lines 180-189, 212, 228, 241, 259, 280, 293, 314, 328, and 335, add information which post hoc test was used to show the statistical significance between the optional two groups.
- Lines 214 and 217, “F4/80+” should be replaced with “F4/80-positive”.
- Line 249, “18-month-old” should be replaced with “12- and 18-month old”.
- Figure 4A, is it possible to add western blotting images of gammaH2AX and p21?
- Figure 7B, is it possible to IHC images of NRF2?
- Figure 7C, is it possible to add western blotting images of phospho-NRF2? IHC images cannot show internal control protein.
- Figure 8, why was the Hmox1 mRNA increased only in 18-month-old mice despite the NRF2 pathway was already the kidney in 12-month-old mice (Figure 7)?
- Lines 387 and 475, “C57Bl6” should be replaced with “C57BL/6”. Please use the same word throughout the manuscript.
- Line 433, “Are” should be replaced with “antioxidant responsive element (ARE)”.
- Please unify the genes symbols according to MGI-guidelines. For example, catalase and p21 are Cat and Cdkn1A, respectively.
Author Response

(The authors gave the same response as above.)

Round 2
Reviewer 1 Report
Authors addressed all my issues
Author Response
Thank you very much for your comments.
Reviewer 2 Report
In the revised manuscript, the authors modified it primarily according to the suggestions by the reviewer. However, only one thing is left for requiring further improvement of the revised manuscript.
Lines 313-314 and lines 537-546, the authors want to show the involvement of immunosenescent cells in the progression of CKD. However, the authors show only the increase of p21-positive cells at 18 months in Figures 4A and 4C. They might reinforce the existence of p21-positive cells with interstitial nuclei (marked by the red arrow). However, Figure 4C did not show the increase of p21-positive infiltrated cells. Therefore, these sentences do not make sense. Please delete these sentences from the manuscript. Instead, the reviewer recommends the authors discuss the possibility of the existence of immunosenescent cells in the 18-month old mice kidney as a limitation of the study. The authors can also add comments on their further study to find the immunosenescent cells in 18 month-old mice kidneys and their involvement in CKD progression.
Author Response
As the reviewer has suggested we have eliminated the data about immunosenescent cells in our study of aging mice.
In particular, regarding Lines 313-314 we have rewritten these lines to better explain our findings, showing that only at 18 months old p21 is significantly increased (at gene and protein levels, evaluated by WB and IH). The IH data only shows the presence of p21 positive cells in the kidney of aging mice, mainly in tubular cells as well as in interstitial cells. To the point of figure C. This figure shows the quantification of gH2AX and p21 positive nuclei per field (this means of all positive cell types).
As indicated additional experiments are needed to identify the cell type of interstitial cells (however this point is beyond the present study).
Therefore (regarding lines 537-546), we only indicate that there is an association of p21 positive interstitial cells and increased infiltration of inflammatory cells in the kidney of aging mice, remarking the importance of future research (limitation).